# Feeding Strategy of the Wild Korean Seahorse (*Hippocampus haema*)

**Myung-Joon Kim** [1], **Hyun-Woo Kim** [2], **Soo-Rin Lee** [3], **Na-Yeong Kim** [1], **Yoon-Ji Lee** [1], **Hui-Tae Joo** [4], **Seok-Nam Kwak** [5] and **Sang-Heon Lee** [1,*]

[1] Department of Oceanography, Pusan National University, Geumjeong-gu, Busan 46241, Korea; mjune@pusan.ac.kr (M.-J.K.); knayo@pusan.ac.kr (N.-Y.K.); yoonji051@pusan.ac.kr (Y.-J.L.)

[2] Department of Marine Biology, Pukyong National University, Nam-gu, Busan 48513, Korea; kimhw@pknu.ac.kr

[3] Industry 4.0 Convergence Bionics Engineering, Pukyong National University, Nam-gu, Busan 48513, Korea; srlee090@pukyong.ac.kr

[4] Oceanic Climate & Ecology Research Division, National Institute of Fisheries Science, Gijang-gun, Busan 46083, Korea; huitae@korea.kr

[5] Environ-Ecological Engineering Institute Company Limited, Haeundae-gu, Busan 48058, Korea; seoknam@eeei.kr

\* Correspondence: sanglee@pusan.ac.kr

**Abstract:** The feeding and spawning grounds for seahorses have been lost due to nationwide coastal developments in South Korea. However, little information on the feeding ecology of the Korean seahorse (*Hippocampus haema*) is currently available. The main objective in this study was to understand the feeding strategy of *H. haema* on the basis of DNA analysis of the contents of the guts. This is the first study on the feeding ecology of *H. haema*. Crustaceans were found to be major prey for *H. haema* in this study. Among the 12 identified species, arthropods were predominantly observed as potential prey of *H. haema* in this study. The *Caprella* sp. Was detected in all summer specimens followed by the *Ianiropsis* sp., whereas isopods were dominant, and amphipods accounted for a small proportion in winter specimens. According to the results in this study, there appears to be a seasonal shift in the major prey of *H. haema*. Moreover, a potential change in the habitats for adults was further discussed. Since this is a pilot study, further studies should be conducted for a better understanding of the feeding ecology of *H. haema*.

**Keywords:** wild seahorse; *H. haema*; feeding habits; NGS analysis

## 1. Introduction

Seahorses are fascinating creatures for many people around the world due to their unique appearance and life-history characteristics that are different to those of common fish [1,2]. The overfishing of seahorses is occurring in some regions [3,4], and reckless coastal developments causing their habitat loss are also threatening their survival [2,5,6]. For this reason, the scientific community is making various efforts to reduce the loss of seahorse populations and preserve the *Hippocampus* species by adding them to lists or conventions, such as the red list of the IUCN (International Union for Conservation of Nature) and CITES (Convention on International Trade in Endangered Species of Wild Fauna and Flora). Until now, there is no record of seahorse overfishing along the Republic of Korea coastal area; however, the Korean coastal ecosystem, due to ongoing nationwide coastal development, has been losing its natural shelter abilities as a safe habitat and spawning ground for various fishes, including seahorses [2,7]. Despite these crises, very little ecological information on seahorses is currently available in the Republic of Korea [8,9]; thus, we do not know what ecological roles they play in coastal ecosystems.

Seahorses lead a sedentary life in coastal areas where the environment can be changed dynamically compared to the open ocean [10–13]. Five seahorse species have been reported

in South Korean waters, mainly in the seaweed and seagrass beds throughout the southern coast [8,9,14–17]. Recently, *H. haema*, previously known as *H. coronatus*, was newly identified as a Korean species [18]. The first study on the morphometric characteristics and basic ecological data for the newly recorded Korean seahorse (*H. haema*) populations was conducted in Geoje-Hansan Bay [8]. However, no study has been conducted on their feeding ecology to date.

The study of the feeding habits of an organism is a cornerstone that can be used for efficient management, preservation, and artificial reproduction of specific organisms, and it is widely conducted for various fish [19–24]. Although several previous studies were conducted on the feeding habits of some seahorse species under laboratory conditions (*H. abdominalis* [25], *H. barbouri* [26], *H. erectus* [27], *H. guttulatus* [28], *H. hippocampus* [29], *H. kuda* [30], and *H. reidi* [31,32]), the information on wild seahorses is still insufficient [24,33–42].

A visual analysis (naked eye or microscope) of gut contents, often used in feeding ecology, can show what quantity of prey is present in the target fish's gut; however, this approach may overestimate some preys that have just been recently eaten or are difficult to digest. Often times, it is difficult to identify what kinds of prey they have eaten due to their easily digestible characteristics or their small size [39,42]. This difference in the digestibility of the prey items can cause some bias when evaluating their diet compositions [39,42]. Stable isotope analysis has the advantage of being able to perform nonlethal analysis of the target organism, but it is difficult to detect the recently eaten prey due to the time gaps in the turnover rate of isotopes [43,44]. On the other hand, metabarcoding has the advantage of being able to detect short-term feeding habits with a very small amount of sample, as well as to detect soft and highly digested items, which are not recognizable through morphological identification [42,45]. Moreover, metabarcoding is also a nonlethal method, depending on the sampling method (i.e., fecal sample [42] or flushing method [34,40]). Therefore, in this study, a DNA analysis method was applied to understand the feeding habits of small *Hippocampus* individuals.

In this study, the main objective was to understand the feeding strategy of the Korean seahorse species (*H. haema*) in the coastal environment on the basis of the contents of the guts in summer and winter, using molecular biological approaches. This is the first study on the feeding ecology of the Korean seahorse species (*H. haema*).

## 2. Materials and Methods

### 2.1. Study Area and Samplings

Among the samples collected by the Environ-Ecological Engineering Institute (EEEI) for the investigation of fish biota in the sargassum bed (*Sargassum piluliferum*), seven specimens were received frozen in summer (July 2017) and winter (January 2019) for a comparison of the feeding strategies of *H. haema*. The sampling area was Geoje-Hansan Bay, Republic of Korea (34°48′40″ N; 128°31′03″ E; Figure 1). The mean water depth of the sampling site was approximately 3 m. The two specimens taken from the samples reported in [8] were analyzed to confirm the possibility of feeding differences between each sampling group even in the summer season (Table 1). The body length and wet weight were measured at the home laboratory using a caliper (Mitutoyo, 0.01 mm, Kawasaki, Japan) and a scale (Mettler Toledo, 1 mg, Columbus, OH, USA) according to [46]. After those measurements, all specimens were stored at −80 °C in a freezer and brought to the Marine Ecological Laboratory in Pusan National University for further analysis. For all sampled specimens (except the two specimens from 2016), the length–weight relationship was calculated using the following equation:

$$Wt = a \times L^b,$$

where *Wt* is wet weight (g), *a* is the intercept, *b* is the slope, and *L* is the standard length (*SL* = straight line between snout and tail, mm). A *t*-test was conducted to verify the statistical difference between the measured values (*SL* and *Wt*) of the January and July groups (SPSS, version 12.0, Chicago, IL, USA). The two specimens in July 2016 were

deliberately selected as relatively large fish; hence, they were excluded from the average and statistical analysis (for a simple DNA comparison).

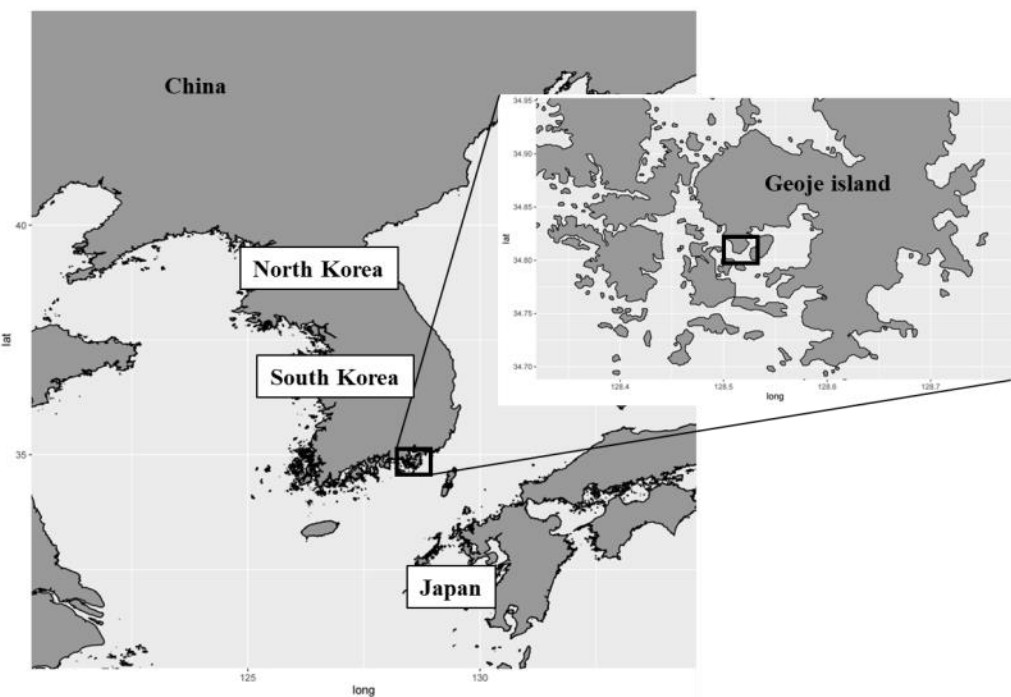

**Figure 1.** Map of the study area in Geoje-Hansan Bay, Republic of Korea (South Korea).

**Table 1.** Measurements of seahorses at each collection period and the *p*-value of the *t*-test.

| Sampling Date | | July 2017 | January 2019 | July 2016 |
|---|---|---|---|---|
| | *n* | 7 | 7 | 2 |
| Range | *SL* | 35.5–89.4 | 52.7–71.1 | 71.4–83.6 |
| | *Wt* | 0.078–1.500 | 0.300–0.721 | 0.681–0.814 |
| Average | *SL* | 58.43 | 60.08 | 77.50 |
| | *Wt* | 0.50 | 0.42 | 0.75 |
| SD | *SL* | 17.27 | 5.96 | 8.57 |
| | *Wt* | 0.49 | 0.15 | 0.09 |
| *t* | *SL* | | 0.815 | |
| | *Wt* | | 0.679 | |

*2.2. Genomic DNA Extraction and NGS Library Construction*

The genomic DNA of *H. haema* was extracted after complete homogenization with Tissue Lyser™ II (Qiagen, Hilden, Germany), by adding tissue lysis buffer to the gut of each of the 16 specimens six times, according to the instructions of the AccuPrep® Genomic DNA Extraction Kit (Bioneer, Daedeok-gu, Korea). ND-1000 (Thermo Scientific, Waltham, MA, USA) was used for the assay and quantification.

The NGS library of the gut contents of *H. haema* was constructed using primers (COIMISQ and NEXCOIMISQ, Table 2) targeting COI in the mitochondrial DNA region, which was previously used for the diet study of *Dissostichus mawsoni* [47]. The blocking primer (Table 2) was prepared the region of the universal primer COIMISQF1 in the nucleotide sequences of *Amphipoda*, *Copepoda*, *Mysidacea*, and *Isopoda*, known as prey organisms, and modified with a C3 spacer at the 3′ end to suppress annealing of the *H. haema* sequence [48]. For the first PCR reaction, a final volume of 25 μL mixed solution

consisting of 10 ng of genomic DNA, 100 μM of COIMISQ primers, 5 μL of 100 μM Blocking primer, 2 μL of 10 mM dNTPs (Takara Bio Inc., Kusatsu, Japan), 0.2 μL of Ex Taq Hot Start Version (Takara Bio Inc., Kusatsu, Japan), 2 μL of 10× Ex Taq buffer (Takara Bio Inc., Kusatsu, Japan), 3% DMSO, and distilled water was used. The reaction conditions and process of the first PCR were as follows: initial denaturation at 94 °C for 5 min, followed by 94 °C, 48 °C, and 72 °C for 30 s, respectively, repeated for 15 cycles. The final extension was conducted at 72 °C for 5 min. Prior to the second PCR reaction, the amplicons were purified using an AccuPrep® PCR Purification Kit (Bioneer, Republic of Korea). For the second PCR reaction, a final volume of 20 μL mixed solution consisting of 5 μL of the purified amplicons, 100 μM of NEXCOIMISQ primers, 2 μL of 100 μM Blocking primer, 2 μL of 10 mM dNTPs (Takara Bio Inc., Kusatsu, Japan), 0.2 μL of Ex Taq Hot Start Version (Takara Bio Inc., Kusatsu, Japan), 2 μL of 10× Ex Taq buffer (Takara Bio lnc., Kusatsu, Japan), and distilled water was used. The reacting conditions were repeated for 18 cycles under the same conditions as that of the first PCR. The amplicons of the second PCR were identified by 1.5% agarose gel electrophoresis (630 bp) and purified by the AccuPrep® PCR Purification Kit. The libraries constructed by the Nextera XT index Kit (Illumina, San Diego, CA, USA) were quantified with a QuantiFluor® Fluorometer (Promega, Madison, WI, USA), before using the MiSeq platform (Illunima, San Diego, CA, USA) for NGS analysis.

**Table 2.** Primers used in this study.

| Name | Direction | Sequence (5′ to 3′) | Reference |
|---|---|---|---|
| COIMISQ | Forward | ATNGGNGGNTTYGGNAA | [47] |
|  | Reverse | TANACYTCNGGRTGNCC |  |
| NEXCOIMISQ | Forward | TCGTCGGCAGCGTCAGATGTGTATAAGAGACAGGGNGGNTTYGGNAAYTG |  |
|  | Reverse | GTCTCGTGGGCTCGGAGATGTGTATAAGAGACAGGGRTGNCCRAARAAYCA |  |
| Blocking Primer |  | GCTTTGGTAATTGACTTG-C3 spacer | This study |

### 2.3. Bioinformatic Analysis

Some parts of the sequence data of the NGS analysis, which featured a short length (less than 100 bp) and low quality (QV < 20), were disregarded. According to [49], the merged reads (more than 470 bp) constructed with the Mothur software package v1.41.3 were used. After the primer sequences were elided, the OTU (Operational Taxonomic Unit) clustering was conducted with Usearch v8.1.1861 [50] at 97%, and the chimeras were discarded. The OTUs were classified as species (≥99%), genus (<99%, ≥90%), and unknown (<90%) on the NCBI GenBank database. Three specimens (No. 2 of 2017, Nos. 4 and 5 of 2019) were excluded from further analysis because no potential prey organisms were identified. The MEGA X Maximum Likelihood method was used for phylogenetic analysis [51].

## 3. Results

### 3.1. Morphometric Measurements

Summer and winter mean surface water temperatures are 24.9 °C and 9.3 °C, respectively, in the study area. A total of 16 seahorses were captured for this study. Table 1 shows the mean *SL* (mm) and *Wt* (g), as well as the results of the *t*-test for each collection period. Although no significant difference in mean *SL* and *Wt* for each collection period was observed, the July specimens had broader ranges in *SL* and *Wt* than the January specimens in this study. The length–weight relationship of each period's specimens is shown in Figure 2. The *b* values (slopes in Figure 2) of the July and January specimens were 3.237 and 3.034, respectively, which were greater than 3, indicating positive allometric growth.

### 3.2. NGS Analysis

As a result of the DNA analysis from the guts of 16 individuals, all specimens were identified as *H. haema*. Of the 2,263,757 reads (mean of 174,135 reads per sample) identified by sequencing, 41.87% were identified as seahorse and bacteria. These reads were excluded from further analysis.

Thus, 47.4% of the nucleotide sequences of the analyzed prey organisms were identified to the lowest taxonomic level (species). Among the 12 identified taxa, 10 were arthropods, and the remainder were *Bryozoa* and fish (Table 3). Among the 10 arthropods, four orders (*Harpacticoida*, *Caprella*, *Ianiropsis*, and *Mysida*) showed a relatively high ratio of over 80% in gut contents of some specimens. In July 2016 and 2017, *Caprella* sp. (amphipods) was detected in all specimens, especially for No. 1 (84.35%) and No. 7 (67.37%), whereas *Mysida* was mostly found in two of the total specimens. The *Ianiropsis* sp. (isopods) was the second most common species. On the other hand, in the January 2019 specimens, isopods tended to be dominant in their prey items, with amphipods accounting for a small proportion (Table 3).

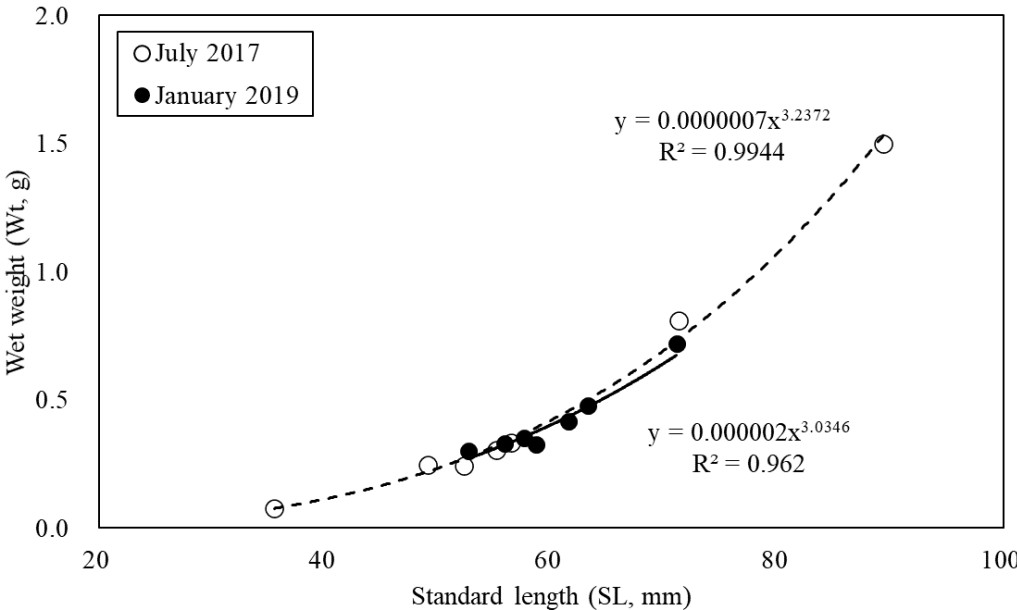

**Figure 2.** Length–weight relationship between specimens collected for each period. The white circles were taken in July 2017, and the black circles were taken in January 2019.

**Table 3.** The results of genetic analysis of gut (up to the species level). Bold font indicates the results of further analysis up to the order level for the results showing "unknown" in the analysis up to the species level.

| | | | | Proportion (%) | | | | | | | | | | | |
|---|---|---|---|---|---|---|---|---|---|---|---|---|---|---|---|
| | | | | July 2016 | | | July 2017 | | | | | January 2019 | | | |
| Phylum | Class | Order | Species | No. 11 | No. 12 | No. 1 | No. 3 | No. 4 | No. 5 | No. 6 | No. 7 | No. 1 | No. 2 | No. 3 | No. 6 | No. 7 |
| *Arthropoda* | *Hexanauplia* | *Calanoida* | *Pseudodiaptomus* sp. | | 0.02 | | | | | | 0.00 | | | | 0.00 | |
| | | **Harpacticoida** | | **43.67** | **97.76** | | **6.67** | | | | | | | | | |
| | | *Harpacticoida* | | 0.13 | 0.12 | | | | | | | | | | | |
| | | | *Amonardia* sp. | 0.07 | | | | | | | | | | | | |

**Table 3.** *Cont.*

| Phylum | Class | Order | Species | Proportion (%) | | | | | | | | | | | | |
| | | | | July 2016 | | July 2017 | | | | | | January 2019 | | | | |
| | | | | No. 11 | No. 12 | No. 1 | No. 3 | No. 4 | No. 5 | No. 6 | No. 7 | No. 1 | No. 2 | No. 3 | No. 6 | No. 7 |
| | | ***Amphipoda*** | | | | | 57.47 | 58.53 | 96.20 | 10.80 | | | 0.14 | | | 0.40 |
| | | *Amphipoda* | *Caprella* sp. | 2.12 | | 84.35 | 9.33 | 21.58 | 3.37 | | 67.37 | | 3.88 | 0.15 | | 0.13 |
| | | *Amphipoda* | *Crassicorophium crassicorne* | | | | | 0.13 | | | | | | | | |
| | | *Amphipoda* | *Gammaridae* sp. | 22.03 | | | | | | | | | 2.90 | | | |
| | *Malacostraca* | *Amphipoda* | *Leucothoe nagatai* | | | | | | | | | | 0.01 | | | |
| | | *Amphipoda* | *Monocorophium acherusicum* | | | | | | | | | | | | | 0.00 |
| | | ***Isopoda*** | | | | | | 7.89 | | | | | | | 13.41 | |
| | | *Isopoda* | *Ianiropsis epilittoralis* | | | | | | | | | | 92.03 | 97.74 | | |
| | | *Isopoda* | *Ianiropsis* sp. | 9.44 | | | | 11.87 | | | | 88.67 | | | | 99.39 |
| | | *Isopoda* | *Munna japonica* | | | | 15.08 | 0.00 | | | 0.00 | | | 0.63 | | 0.01 |
| | | ***Mysida*** | | | | | | | | 89.02 | | | | | 86.59 | |
| *Bryozoa* | *Gymnolaemata* | *Ctenostomatida* | *Amathia verticillate* | 0.17 | | 0.26 | 0.04 | | 0.06 | 0.00 | 0.08 | | | | | |
| *Chordata* | *Actinopterygii* | *Perciformes* | *Pictichromis paccagnellae* | | | | | | 0.23 | | | | | | | |
| | | | Unknown | 22.36 | 2.10 | 15.39 | 11.42 | | 0.15 | 0.18 | 32.55 | 11.33 | 1.04 | 1.48 | | 0.07 |
| | | | Total | 100 | 100 | 100 | 100 | 100 | 100 | 100 | 100 | 100 | 100 | 100 | 100 | 100 |

Since each specimen could be different in their energy budget requirements according to their body size (growth), the seahorse's diet would be different by size in the summer with various sizes of seahorses. However, among the specimens collected in July 2017, the largest seahorse (No. 1, *SL*: 89.4 mm) had a higher proportion of *Caprella* sp. (84.35%) at the species level, while the smallest one (No. 5, *SL*: 35.5 mm) had mostly Amphipoda (96.2%) at the order level (Table 3). In both cases, the Amphipoda order showed a tendency to occupy the highest ratio, although it was not possible to compare them due to the difference in the lowest taxonomic level of identification.

As a result of additional analysis of the 52.6% classified as "Unknown", clustering was confirmed at the order level (Figure 3). Because Figure 3 does not indicate the proportion of gut contents but just all items identified up to the order level, some items with very low percentages (e.g., Algae, Calanoida, Ctenostomatida, and Perciformes) are not shown in Table 3 or Figure 4a. The difference in the gut composition in each collection period was clear in comparison at the order level rather than the species level (Figure 4a). The proportions of Harpacticoida and Amphipoda had high ratios in July 2016 and 2017, respectively, while Isopoda was dominant in January 2019. This result was consistent with the similarity analysis in this study (Figure 4b).

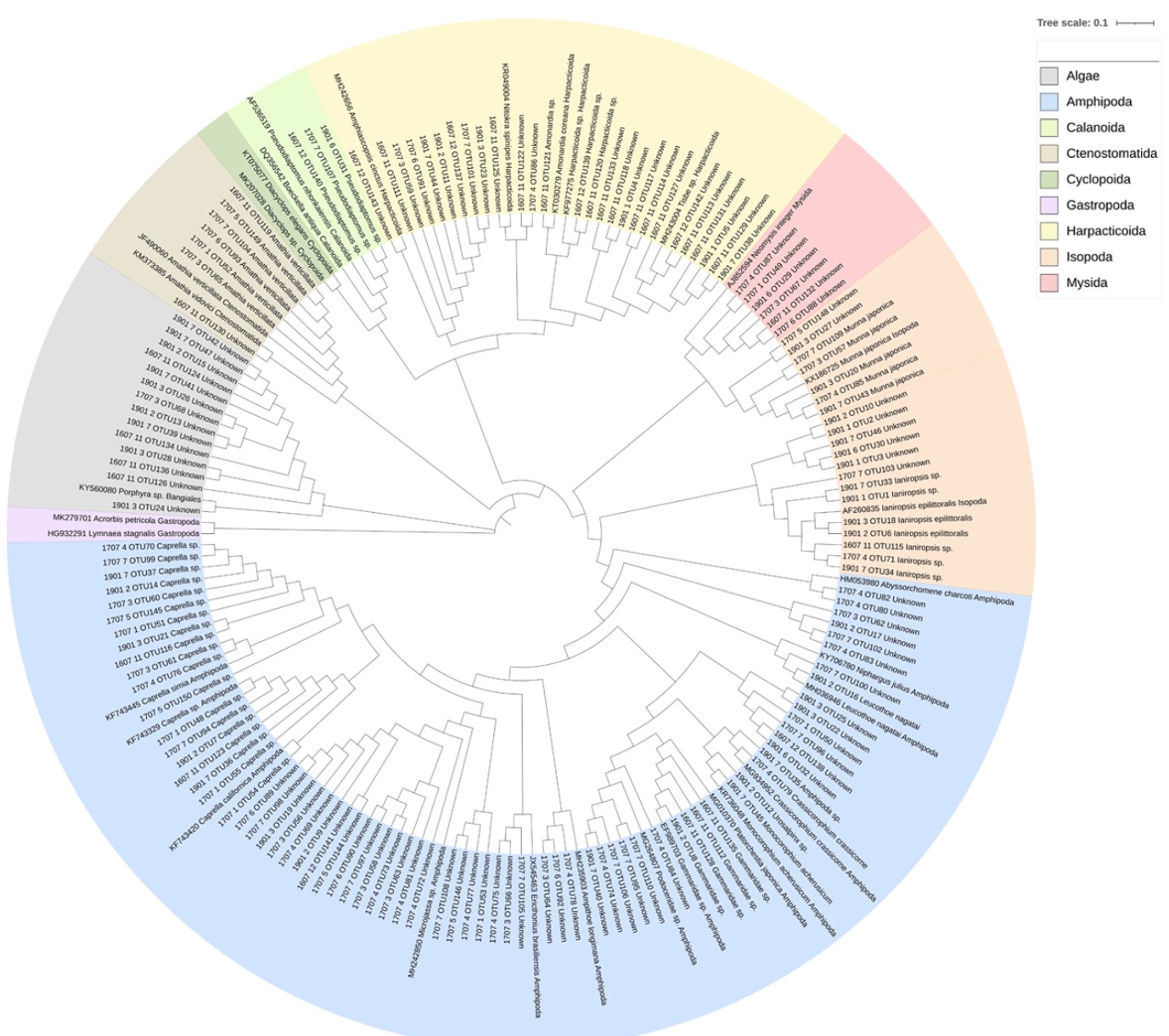

**Figure 3.** Phylogenetic tree of gut contents.

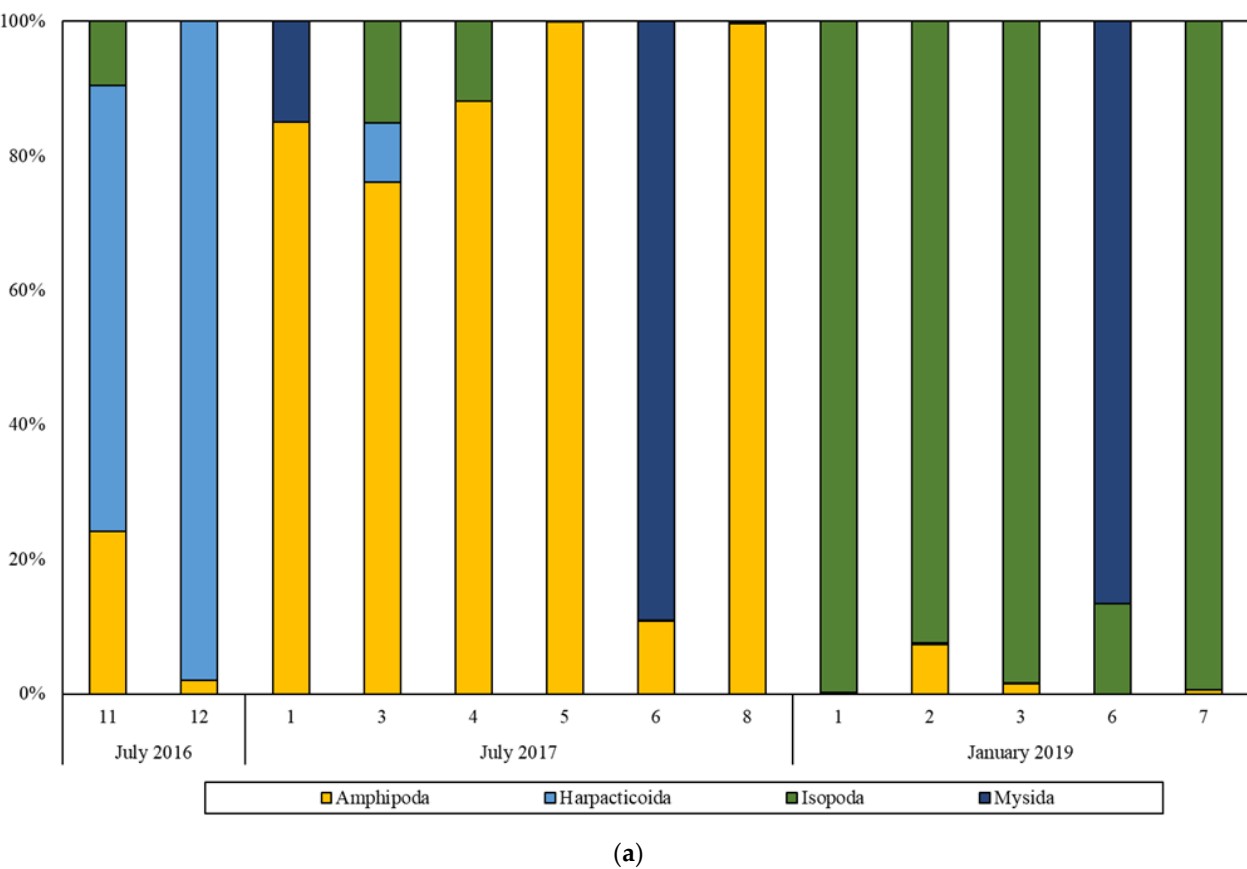

(**a**)

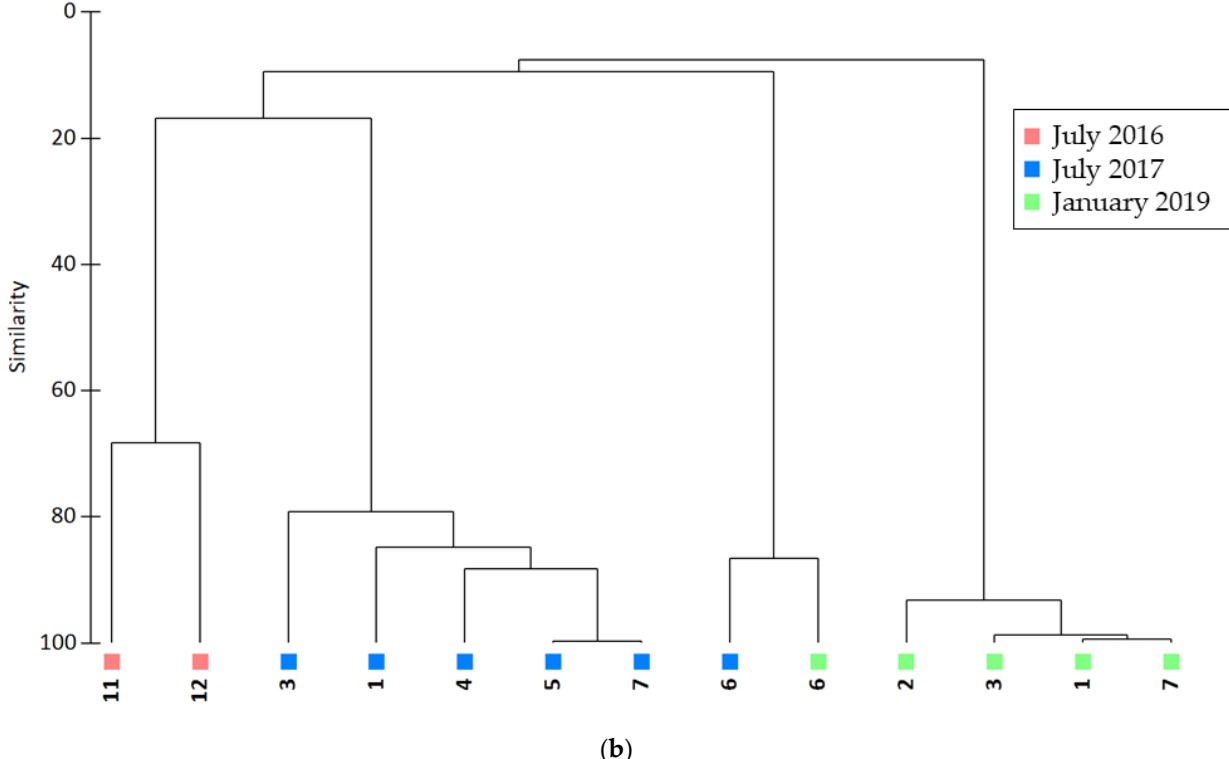

(**b**)

**Figure 4.** (**a**) The order-level taxonomic ratio of gut contents analyzed with COI primer. (**b**) Similarity analysis of gut contents by collecting period.

## 4. Discussion

To compare the seasonal feeding habits of *H. haema* during summer and winter, 14 specimens were collected from Geoje Hansan Bay in July (seven) and January (seven), when the seasonal characteristics were distinct. All the specimens were found in *Sargassum piluliferum* [8] and identified as the same species by DNA analysis [18]. There was no significant difference in mean *SL* and *Wt* by the two different seasons, but there was a relatively wider range in the mean *SL* and *Wt* in the July specimens. In other words, more varied sizes of seahorses were captured in July compared to January. According to previous studies, the breeding season of seahorse is from late spring to autumn in Korea [8,9], and there is a positive correlation between the growth of the genus *Hippocampus* (*H. whitei*, *H. guttulatus*, and *H. zosterae*) and water temperature within a certain range [13,52,53]. Thus, the appearance of seahorses of various sizes in July could be due to the recruitment of newborns and the relatively fast growth rates during the summertime.

*NGS Analysis*

A total of 12 taxa were identified in all gut contents except for three specimens without genetic information for any prey items. In comparison with other previous studies on the seahorse diet [24,34–42], Bryozoa was detected in a small proportion in this study, which might indicate that bryozoans were fed non-selectively with other prey.

Fish was detected only in one individual in this study. We presumed that other fish or eggs were misidentified due to the lack of NCBI data. *Pictichromis paccagnellae* is a tropical species which does not live in Korean waters. Although it is also assumed to be misidentified, it is likely that the genetic information could be other fish larvae or eggs, as actual feeding larvae have been reported in several different kinds of seahorses such as *H. abdominalis*, *H. reidi*, and *H. trimaculatus* [24,34,38].

Some "unknown" genetic information, which could not be confirmed even at the order level, suggests that there are still insufficient coastal zooplankton DNA library data for a variety of zooplankton in coastal areas.

Like other seahorse species reported previously [3,23,24,26,33–42,54–56], *H. haema* appears to feed mainly on crustaceans, according to the results in this study. However, the gut contents of *H. haema* were clearly differentiated by different sampling periods at the order level as shown in Figure 4a,b. *Caprella* was detected in most specimens in this study regardless of season, but a large proportion was recorded mainly in summer. In contrast, *Ianiropsis* accounted for a large proportion of gut contents in winter, showing a pattern opposite to that of *Caprella*. In fact, although *Caprella* appears in abundance on the coastal areas all year round [57–59], it is most often found in summer due to its strong tolerance to high water temperatures [57]. In comparison, large numbers of *Ianiropsis* are normally observed in winter [60].

Mysida species are generally observed throughout the year in coastal waters, which have an extreme seasonal temperature difference of more than 20 °C [61,62]. They stay and/or migrate in several swarms on the sandy bottom, near vegetation and rocks [61]. In general, seahorses forage accessible prey (slow and smaller than their mouths) within vegetation, but they occasionally hunt fast prey such as Mysida and Caridea on sandy bottoms [24,34]. The fact that Mysida was found only in some seahorses in this study indicates that several active individuals actually attempted to hunt them when the Mysida swarm reached their vicinity, but it was not easy for the seahorses to catch them due to the rapid reaction rate of Mysida. This can be supported by the low hunting success rates of *H. haema* toward a Mysida swarm observed in a breeding water tank (unpublished data).

Although Harpacticoida accounted for a large percentage of the specimens in July 2016, they occupied a very small portion for the prey items for *H. haema* in July 2017. Normally, Harpacticoids are a major item for some *Hippocampus* species (*H. zosterae*, [23]; *H. reidi*, [34]; and *H. subelongatus*, [54]). The seahorses can change their feeding capacity as they grow [3,23,24,63]. Indeed, in a laboratory environment, copepods (<1 mm) were eaten by all *H. haema* regardless of their size, whereas Artemia and Mysida (>1 mm) were

mainly eaten by large seahorses (unpublished data). Like *Caprella*, harpacticoids appear frequently in the late spring–summer period [64,65], which coincides with the breeding season of *H. haema*. Therefore, with a maximum size of approximately 1 mm, harpacticoids could be an important food source for growing seahorses with their small mouth sizes.

The coincidence between the gut contents of *H. haema* and the seasonally thriving zooplankton in our study indicates that the main diet of *H. haema* can be modified according to the availability of prey. The authors of [24] found that wild *H. abdominalis* collected from Wellington Harbor did not have a seasonal shift in their habitats, but there was a major seasonal change in their prey. According to a previous study that analyzed feces from *H. guttulatus* using a genetic method, the seahorse diet differs depending on the habitat [42]. Although some differences among species are expected, this ability of *Hippocampus* species to flexibly change their diets in response to spatiotemporal changes in their surroundings could be one of the survival strategies for adapting to dynamic coastal environments.

## 5. Conclusions

Although specimens were collected once each in summer and winter, differences in feeding habits of *H. haema* could be clearly distinguished using DNA tools. However, due to the lack of genetic information on coastal zooplankton, it was difficult to identify up to the species level what seahorses mainly feed on. If coastal zooplankton can be identified up to the species level and their ecological characteristics understood, it could make a great contribution to understanding the role of the seahorse from an ecological perspective. In conclusion, this study is very important to provide the first information on the feeding behaviors of the Korean seahorse species, which is very valuable for their sustainable management and conservation in Korea.

**Author Contributions:** M.-J.K. and S.-H.L. conceived and designed the experiments; H.-W.K. and S.-R.L. performed the experiments; M.-J.K. and S.-R.L. analyzed the data; S.-H.L. and H.-W.K. validated the results; M.-J.K., N.-Y.K., Y.-J.L. and S-N.K. investigated; M.-J.K., N.-Y.K. and Y.-J.L. curated the data; M.-J.K. wrote the original draft; M.-J.K. and S.-H.L. reviewed and edited the manuscript; S.-N.K. and H.-T.J. visualized the data; S.-H.L. supervised this research; S.-H.L. and H.-T.J. funded acquisition. All authors have read and agreed to the published version of the manuscript.

**Funding:** This work was supported by a 2 year research grant from Pusan National University.

**Institutional Review Board Statement:** Ethical review and approval were waived for this study due to all fish samples were dead when we received them.

**Informed Consent Statement:** Not applicable.

**Data Availability Statement:** Not applicable.

**Acknowledgments:** The authors would like to thank the anonymous reviewers and the handling editors who dedicated their time to providing the authors with constructive and valuable recommendations.

**Conflicts of Interest:** The authors declare no conflict of interest.

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
