# Peer review of "Feeding Strategy of the Wild Korean Seahorse (Hippocampus haema)"

_jmse, doi:10.3390/jmse10030357_

Round 1

Reviewer 1 Report

Dear editor,

Thank you for providing me this chance to review this paper. I have read the paper carefully and offer some comments and suggestion.

In brief, I think this paper is proper and good enough for publication in your journal, except that the authors need to revise the MS carefully. Especially, the methods are still not very clear for the authors.

Best,

Dr, Geng Qin, Associate Professor

South China Sea Institute of Oceanography, SCSIO.

Reviewer 2 Report

Dear Authors in the attache files my suggestions and comments.

I hope my suggestions can help you to improve the MS quality.

All the best

MS ID: jmse-1569403

Kim, M. J.; Kim, H.-W.; Lee, S. R.; Kim, N.-Y.; Lee, Y.; Joo, H.T.; Kwak, S. N.; Lee, S. H. Seasonal feeding strategy of the wild Korean seahorse (Hippcampus heama). J. Mar. Sci. Eng. 2022.

GENERAL COMMENTS

The manuscript provides useful information on the diet of H. heama using Genomic DNA extraction and NGS library construction. The study provides useful data on the on wild diet of H. heama in two periods of the year (winter and spring). The species is little studied, so all information on its biology and ecology is welcome and deserving of publication. However, before being considered for publication several aspects need insights and clarifications.

  • I would recommend that the Authors re-check and improve the English of the manuscript. I am not a native speaker but I believe the text needs a substantial revision. In the following pages I have pointed out some minor errors but English, in general, does not sound good.
  • Introduction should be improved. First check your English; then, the authors should better explain why they used " the DNA analysis”, cite appropriate references and contextualize it in the context of diet studies.
  • Mat and Met are unclear and need tidyng up. I did not understand who captured the animals and for what purpose. The habitat in which they were sighted is not indicated in Mat and Met (but indicated in discussion at line 181 - Sargassum piluliferum).
  • Discussion should be aligned with the introductions and the authors should essentially discuss the results of the work without making too much speculation.

In the following pages several comments and suggestions poin by point.

Moreover in the attached pdf file I have highlighted further weaknesses.

Title

Seasonal feeding strategy of the wild Korean seahorse (Hippcampus heama).

The sampling conducted for this study is not seasonal (four seasons) but biannual (summer and winter).

Ethis statement

In the Ethics statement (lines 280-281) the Authors write “Ethical review and approval were waivered for this study because all fish samples were dead when we received them”. However  in “Study area and samplings” paragraph the authors write “To reduce some potential damage for the seahorse population, a few specimens as possible were collected by hand-picking without using any fishing tool.

Please explain better.

INTRODUCTION

Line 31: life-historical -> life-history.

Line 32: I suggest changing ordinary to common.

Line 34: human beings are making; scientific community beings are making

Line 35: of their population -> of seahorse populations

Line 35: hippocampal; in my opinion it is a term used in human anatomy.

Line 35: treats ???

Lines 37-40: this sentence is uncleare and a little confused. Please rewrite.

Results

In Fig. 1 seems that alge represent one important item; however this”prey” is missing in Tab.1.

Tab 1 The authors worked on a wide range of SL (above all in 2017 35.5/89.4). I believe that between these specimens there can be great differences in terms of the energy budget required and the size of the prey

Discussion

Line 197: H. zosteae -> H. zosterae

Lines 190-202: The authors discuss at length the possible winter migration of adult specimens to greater depths. It doesn't seem like the focus of the study to me I suggest staying focused on scopes and aims of the ms and only discussing the results obtained in the study

Lines 202-204: Authors should cite some bibliographic references on this topic; otherwise, it remains too speculative.

Line 225- shown in Figures 4 and 5. Maybe Figures 4a and 4b!!

Reviewer 3 Report

This study aims to determine the feeding ecology of a poorly studied seahorse species (H. haema). Although the study is relevant, there are some issues that have to be tackled before the manuscript is ready for publication. My major concern is the fact that one of the rationale given to support the validity of the study (i.e. "This study could provide valuable information for sustainable management and conservation for the Korean seahorse species") is not sufficiently discussed in the Discussion section. There are a few speculative statements that need to be supported by references to other studies or removed from the text. The manuscript would highly benefit from a revision by an English native speaker.

Round 2

Reviewer 2 Report

MS ID: jmse-1569403 – Vs 2

Kim, M. J.; Kim, H.-W.; Lee, S. R.; Kim, N.-Y.; Lee, Y.; Joo, H.T.; Kwak, S. N.; Lee, S. H. Biannual feeding strategy of the wild Korean seahorse (Hippcampus heama). J. Mar. Sci. Eng. 2022.

Title - Biannual feeding strategy of the wild Korean seahorse (Hippocampus haema)

In the previous version of the MS I have emphasized  that your sampling strategy was not Seasonal but biannual. However I don't like the title in its present form. I suggest to completely delete “biannual” and leave alone “Feeding strategy of the wild Korean seahorse (Hippcampus heama)”. You can specify the sampling strategy in the Mat and Met. What do you think about it?

Abstract

Line 17 –seasonal again!

Lines 37-40 - Fortunately, there is no record of overfishing of seahorse yet; however, the coastal ecosystem in the Republic of Korea has been losing its natural shelter abilities as a safe habitat and spawning grounds for various fishes, including seahorses [2, 7].

This sentence doesn't sound good. I propose you a possible alternative.

Until now, there is no record of seahorses overfishing along the Republic of Korea coastal area/ecosystem/system; However, the Korean coastal ecosystem, due to ongoing nationwide coastal development, has been losing its natural shelter abilities as a safe habitat and spawning grounds for various fishes, including seahorses [2, 7].

Lines 90-91 - All specimens except the two specimens from 2016 were calculated using length-weight relationship equation.

It is not the individuals that are calculated but the length-weight relationship.

For all sampled specimens (except the two specimens from 2016) length-weight relationship was calculated using the following equation:…..

Lines 90-91 - Among the 12 identified species, 10 were arthropods, and the remainder were Bryozoa and fish (Table 3).

In my opinion the authors have identified 12 taxa (not species) of which eleven at the species level.

Tab. 3 - The numbers in the third line of the table (N° 11, N° 12, .. etc) appear to be a unique identification code for Seahorse specimens. Why are there only 13 specimens? Did you write somewhere that three specimens were not included in the analysis? Sorry, but I can't find this sentence.

In the attached pdf file You can find further comments.

All the best
